# Fexinidazole for Human African Trypanosomiasis, the Fruit of a Successful Public-Private Partnership

**DOI:** 10.3390/diseases10040090

**Published:** 2022-10-17

**Authors:** Sonja Bernhard, Marcel Kaiser, Christian Burri, Pascal Mäser

**Affiliations:** 1Swiss Tropical and Public Health Institute, 4123 Allschwil, Switzerland; 2University of Basel, 4002 Basel, Switzerland

**Keywords:** human African trypanosomiasis, sleeping sickness, *Trypanosoma brucei*, drug discovery, nitroimidazole, product development partnership, clinical trial operations, clinical trial planning, clinical trial efficiency

## Abstract

After 100 years of chemotherapy with impractical and toxic drugs, an oral cure for human African trypanosomiasis (HAT) is available: Fexinidazole. In this case, we review the history of drug discovery for HAT with special emphasis on the discovery, pre-clinical development, and operational challenges of the clinical trials of fexinidazole. The screening of the Drugs for Neglected Diseases initiative (DNDi) HAT-library by the Swiss TPH had singled out fexinidazole, originally developed by Hoechst (now Sanofi), as the most promising of a series of over 800 nitroimidazoles and related molecules. In cell culture, fexinidazole has an IC_50_ of around 1 µM against *Trypanosoma brucei* and is more than 100-fold less toxic to mammalian cells. In the mouse model, fexinidazole cures both the first, haemolymphatic, and the second, meningoencephalitic stage of the infection, the latter at 100 mg/kg twice daily for 5 days. In patients, the clinical trials managed by DNDi and supported by Swiss TPH mainly conducted in the Democratic Republic of the Congo demonstrated that oral fexinidazole is safe and effective for use against first- and early second-stage sleeping sickness. Based on the positive opinion issued by the European Medicines Agency in 2018, the WHO has released new interim guidelines for the treatment of HAT including fexinidazole as the new therapy for first-stage and non-severe second-stage sleeping sickness caused by *Trypanosoma brucei gambiense* (gHAT). This greatly facilitates the diagnosis and treatment algorithm for gHAT, increasing the attainable coverage and paving the way towards the envisaged goal of zero transmission by 2030.

## 1. A Neglected Disease of Neglected Patients

Deadly epidemics emerging in tropical Africa, a fragmentary surveillance at best, no vaccine, and drugs that are toxic: the outlook for human African trypanosomiasis at the end of the last century was grim indeed. In this case, 60 million people were at risk, half a million estimated to be infected, but fewer than 10% were under surveillance and receiving proper care [1,2]. Human African trypanosomiasis (HAT, a.k.a. sleeping sickness) is a neglected tropical disease caused by two subspecies of *Trypanosoma brucei* that have evolved resistance to human serum: *T. b. gambiense* in West and Central Africa, *T. b. rhodesiense* in East Africa [3]. The parasites are transmitted through the bite of the hematophagous tsetse fly (*Glossina* spp.) [4]. Competent vectors include savannah, sylvatic, and riverine species. Given their peculiar biology, in particular the fact that they are viviparous, the tsetse flies have never established populations outside their natural habitat in tropical Africa; thus, HAT remained a disease of the rural poor, threatening the people at exposed outdoor sites such as river crossings, water collection points, or washing sites. The epidemics in the 1990s were caused mainly by *T. b. gambiense* in the war-ridden regions of Angola, the Democratic Republic of the Congo (DRC), and southern Sudan/northern Uganda [4]. The last *T. b. rhodesiense* outbreaks occurred in eastern Uganda and northern Malawi with cattle and wild mammals, respectively, acting as the potential reservoirs for human-pathogenic trypanosomes [5,6].

## 2. The Chemotherapy of Sleeping Sickness: Old Roots but Little Fruit

Untreated, sleeping sickness usually takes a fatal course once the trypanosomes have crossed the blood-brain barrier and infested the cerebrospinal fluid. Drug discovery for African trypanosomiasis was of great importance to pharmaceutical history. In fact, the first chemotherapeutic studies were carried out with African trypanosomes [7]. Using synthetic dyes produced by the new pharmaceutical industry, Paul Ehrlich had founded the field of rational drug development. This culminated in suramin as a first antitrypanosomal drug, developed by Bayer in 1916 [8]. Pentamidine would follow in the 1930s [9,10]. These drugs did not penetrate the blood-brain barrier, though, and therefore only worked during the first, haemolymphatic stage of the infection [4]. For a long time, the sole cure for the second, meningoencephalitic stage were arsenicals, which, to no surprise, were highly toxic. Ernst Friedheim in 1948 combined melarsen with the arsenic antidote British anti-Lewisite to build melarsorpol [11]. However, the toxicity was still severe: up to 5% of the treated patients developed lethal encephalopathies [12]. Furthermore, to make matters worse, melarsoprol resistance was emerging in *T. b. gambiense* [13,14].

Fortunately, an alternative brain-permeant trypanocide was discovered in the form of eflornithine (difluoromethyl-ornithine, DFMO), an irreversible inhibitor of ornithine decarboxylase [15]. Originally developed as an anticancer drug, eflornithine was repurposed for HAT [16]. The drawbacks of eflornithine were that it was not effective against *T. b. rhodesiense* and that even for *T. b. gambiense*, it required such a high dose as could only be administered by four daily infusions over 14 days [17]. Nevertheless, the new treatment option mainly used by non-governmental organizations, optimized tsetse traps, intensified surveillance and control measures combined with increased awareness, halted the epidemics [18]. In addition, the funding for HAT had increased, in particular thanks to the Bill and Melinda Gates Foundation, and in 2009 the Drugs for Neglected Diseases initiative (DNDi) was able to have NECT (nifurtimox-eflornithine combination therapy [19]) entered to the essential medicines list (EML) of the WHO as a new treatment option [20]. NECT had a shorter regimen than eflornithine monotherapy, requiring 14 rather than 56 infusions. However, it necessitated a complicated diagnostic algorithm involving a lumbar puncture for stage determination, highly trained personnel, dedicated HAT clinics, and sophisticated logistics (see Figure 1).

In parallel, the development of an oral sleeping sickness drug (DB289) for first-stage HAT was carried out by the public private partnership (PPP) Consortium for Parasitic Drug Development (CPDD) [21]. First studies in patients started in 2001, the phase III study was conducted from 2005 to 2009 [22,23]. Unfortunately, the 10 days course of DB289 failed due to late nephrotoxicity at a very late stage in the development in 2008. Therefore, despite the long history of research, the development of in vitro systems for drug screening [24], an increased understanding of the trypanosomes’ peculiar biology [25], and breakthroughs in genomics [26], there was still no drug that would provide a safe cure of HAT and a simple regimen in the mid-2000s. At least, the DB289 program had paved the path for further clinical trials with sleeping sickness drug candidates such as fexinidazole.

## 3. Preclinical Development of Fexinidazole

Encouraged by the steadily declining incidence since the control of the last epidemics, the WHO has envisaged the elimination of HAT—at least of *gambiense* HAT, which does not have the same zoonotic reservoir as *rhodesiense* HAT [27,28]. However, all modelling scenarios pointed out that the minimal treatment coverage required for elimination was achievable only with new drugs [29]. In particular, a safe, oral drug was needed that could be given as a pill.

In addition to screening new chemistry, DNDi had reconsidered also an old chemical class of proven antimicrobial activity and oral bioavailability: the nitroimidazoles (Figure 2). Under the leadership of DNDi over 800 nitroimidazoles had been collected from pharma industry, academic groups, and retired chemists over the whole world. All these molecules were tested according to the algorithm shown in Figure 3, starting with an integrated in vitro screen determining the antiparasitic activity against *T. b. rhodesiense* bloodstream forms, *T. cruzi* intracellular amastigotes, *Leishmania donovani* axenic amastigotes, and cytotoxicity on L6 rat myoblast cells. In this case, 80 molecules were identified as active against *T. b. rhodesiense*, 35 of which with a high selectivity index (SI > 100, where SI equals the IC_50_ to L6 cells divided by the IC_50_ to the parasite). Of these compounds with selective activity, 8 were able to cure the acute mouse model mimicking the first stage of the disease by oral administration. The chronic mouse model mimicking the second stage of the disease with a CNS infection was cured by 4 compounds only. Due to its good in vitro and in vivo activity profile and low genotoxicity, fexinidazole was singled out as the molecule to be pursued. Fexinidazole (Hoe 239) is a 2-substituted 5-nitroimidazole (Figure 4) that had been in preclinical development by Hoechst [30,31]. The original patents for fexinidazole (US4042705A and CA1079738A) have expired [32]. However, there is an active patent (US9585871B2) for the use of fexinidazole for leishmaniasis in dogs [32], now held by Boehringer Ingelheim Animal Health after their acquisition of Merial.

Fexinidazole and its two main metabolites fexinidazole-sulfoxide and -sulfone were active in vitro against a panel of *T. brucei* subspecies including sensitive and resistant lab strains and field isolates (IC_50_ 0.7–3.3 μM) [33,34] (Figure 4). Oral administration of fexinidazole at a dose of 100 mg/kg/day for 4 days cured the acute mouse models infected with *T. b. rhodesiense* (STIB900) or *T. b. gambiense* (130R) [33,34]. Fexinidazole cured the chronic mouse model (*T. b. brucei* GVR35) at an oral dose of 100 mg/kg/bid for 5 days (Figure 4). The high in vivo efficacy of fexinidazole had already been demonstrated by Jennings and Urquhart (1983); they were able to cure 80% of the infected mice with a dose of 250 mg/kg/day given on 4 consecutive days [35]. The bioavailability of fexinidazole is 41% when administered orally (Figure 4), and the compound readily distributes throughout the body, including the brain [34]. Furthermore, the pharmacological and toxicological studies confirmed the safety and good tolerability of fexinidazole [33,34,36].

## 4. Fexinidazole Is a Magic Bomb

Fexinidazole does not fit into the classical picture of the magic bullet, i.e., a drug that precisely hits a specific target inside the parasite which is absent (or sufficiently different) in the host. The nitroimidazoles are more similar to magic bombs [37]. They are prodrugs that are activated only inside the parasite. This activation occurs by chemical reduction, i.e., the acquisition of one or two electrons. Nitroimidazoles and nitrofurans are used against a wide spectrum of pathogens including *T. brucei*, *T. cruzi*, *Entamoeba*, *Giardia*, *Trichomonas vaginalis*, *Bacteroides*, *Clostridium difficile*, *Helicobacter pylori*, and other anaerobic bacteria. These different species are clearly not phylogenetically related; what they have in common is an electron donor of sufficiently low redox potential to reduce the nitro group. In the case of trypanosomatids, this is NADH-dependent nitroreductase I, a mitochondrial enzyme that catalyses the two electron transfer reducing the -NO_2_ group of fexinidazole to nitroso (–N=O), hydroxylamine (-NH-OH), and finally amine (-NH_2_) [38,39]. The two intermediates are highly reactive. Unspecific reaction with biomolecules, including DNA, is what causes the cytotoxic and mutagenic effects of the nitroimidazoles [39].

Nitroreductase I is absent from animals. It is a typical bacterial gene, and it is not clear how it has made it into the genome of the trypanosomatid ancestor, possibly by horizontal transfer. Nitroreductase I genes are present also in the *Salmonella typhimurium* strains used for the Ames test, which explains why the nitroimidazoles all have a positive outcome (and why the chemical industry had lost interest in this class of molecules). However, fexinidazole and its sulfoxide and sulfone metabolite were not mutagenic when tested against *S. typhimurium* nitroreductase loss-of-function mutants, and neither were they mutagenic in micronucleus tests performed with mammalian cells [36]. These findings cleared the way for the clinical development of fexinidazole. Indeed, fexinidazole turned out to be well tolerated with only mild or moderate adverse effects, the most frequent being headache, vomiting, nausea, and insomnia [40,41,42]. More serious effects, in particular signs of liver toxicity, were observed during long-term (e.g., 8 weeks) treatment of Chagas disease patients with fexinidazole [43]. However, the drug exposure reached during the regimen for HAT lies below the safety margin for hepatotoxicity [43].

## 5. Clinical Development of Fexinidazole

HAT is a complex, life-threatening disease that is characterized, particularly in its second stage, by nonspecific neurologic or psychiatric symptoms such as imperative sleep disturbances, headaches, mood or behavioral changes, seizures, abnormal movements, paralysis, and relapses possible for a long time after treatment. This leads to several impediments to clinical trials: frequent serious adverse events (SAE) due to disease severity, the risk of stigmatization of the patients, and a mandatory follow-up period of up to two years. In addition, the fact that HAT occurs in very remote areas that are difficult to reach and deprived in resources results in major challenges: long, difficult and expensive travels to the study sites, research-naïve site staff, rudimentary infrastructure lacking the necessary laboratory equipment, and limited access to communication means.

The clinical development of fexinidazole was initiated in 2009 with altogether three phase I trials. Safety and efficacy of fexinidazole, formulated as 400 mg tablet, against *gambiense* HAT was demonstrated through a pivotal phase II/III trial in second-stage patients and two cohort/plug-in studies in first- and early second-stage patients and in children 6 to 15 years of age of all disease stages [40,41,42]. The study sites were in the DRC and the Central African Republic (Figure 5). The performance of fexinidazole was satisfactory, resulting in a positive scientific opinion by EMA in 2018. Continued exchange between the consortium and the WHO throughout the whole drug development process resulted in new WHO interim guidelines in 2019 [44]. This allowed the affected countries such as DRC and Uganda to almost immediately approve and implement the new treatment. US FDA approval was received in 2021. Fexinidazole is being donated to the WHO by Sanofi, and the WHO then distributes the drug to the National Control Programmes of HAT-endemic countries, where the treatment is available for free [45].

The successful clinical development of fexinidazole was possible thanks to the joint effort of many actors, including DNDi as the initiator and sponsor of the drug development program, National HAT Control Programmes, regulatory authorities, local scientists, site staff, local communities, the WHO, industrial partners including Sanofi, NGOs, and funding agencies. On the mandate from DNDi, Swiss TPH contributed the monitoring and coordination of the phase II/III studies. In the following, we focus on the implemented mitigation activities to the challenges mentioned above.

## 6. Elements of Success

A key element for the success of the fexinidazole development program was the detailed planning of the phase II/III clinical studies. On the one hand, the target-product profile had been precisely defined by the different national and international stakeholders and was therefore well adapted to the needs of patients, practitioners, and countries. On the other hand, the study protocol was elaborated to guarantee the feasibility of such complex research in remote hospitals where the HAT patients are typically treated, and to ensure adherence to the GCP requirements. The value and positive impact of this process is often underestimated. It has been shown that detailed planning of a study and writing of the protocol with participation of the local professionals, and adaptation to the local circumstances, has a very beneficial effect on the overall efficiency and quality of clinical trials in resource-limited settings [46,47,48].

Another important element to any successful study is access to the participants. In the DRC, detailed data and maps on patient location are available at the National Sleeping Sickness Programme [49]. Despite this knowledge, due to the low prevalence of HAT even in disease foci, over two million people had to be screened to enroll the 749 patients [45]. The study sites had to be refurbished and equipped, which included painting, renovation and installation of solar panels and satellite dishes; ECG apparatuses, centrifuges, biochemistry analyzers and pipettes had to be provided at all sites; the equipment had to be calibrated and validated in the absence of certification programs. DNDi ensured the provision not only of the study drugs and reactants, but also of additional consumables and medicines for which stockouts are rather the norm than the exception in such remote health facilities [50].

For data capture in the phase II/III main study, an online solution with additional off-line functionality was used. However, power failures occurred quite frequently impacting communication and data synchronization. Consequently, traditional paper data capture solutions were used for the subsequent plug-in studies. Another threat that had to be mitigated was the high staff turnover common in remote health facilities. Trained staff leaving were generally replaced by staff naïve to research. The knowledge loss was overcome through continued and repeated training. The DNDi local office appointed several professionals specifically to train site staff on laboratory procedures, patient management, waste disposal, hygiene, and investigator responsibilities.

Another key to success was intensive monitoring and continuous coaching/mentoring. Monitoring capacity at that time was limited in the countries where the studies were carried out. Therefore, talented candidates were selected from local life science research institutions and trained to become monitors. All study monitors had a medical or laboratory background and were familiar with sleeping sickness. The training included one basic five-day training, several refresher trainings, study-specific trainings and hands-on training, as well as coaching during monitoring visits that were carried out together with monitors from Swiss TPH. The pivotal study was launched in 2012 and carried out in nine hospitals in DRC and one in the CAR (Figure 5); the cohort studies that were added in 2014 took place in eight of the nine hospitals in the DRC. Prior to any patient recruitment, several visits to pre-screen possible study sites were undertaken, followed by 36 visits to assess the study sites for suitability and prepare them for study commencement. From 2012 to 2018, for all three studies combined, a total of 570 on-site visits were carried out to ensure the quality of the study. From 2014 onward, the visits were often combined to cover all three ongoing studies (Figure 6). During the study conduct phase (October 2012 to May 2018), 476 monitoring and 58 close-out visits were carried out. Almost half of these visits were conducted by local monitors newly trained for this study; during the other site visits local monitors were accompanied by monitors from the Clinical Operations Unit (COU) of Swiss TPH. These co-visits guaranteed a stable quality and contributed to the capacity building to which both, DNDi as well as Swiss TPH, commit in their mandates. Over the whole study period, more than 2600 days were spent on site by a team of 17 monitors.

In several instances, situations of political instability and unrest hampered the monitoring activities. Foresight and flexibility were not only asked for from the sites, but also from the monitoring team. For instance, time-points and duration of monitoring visits had to be adapted to avoid security risks. In other occasions, a specific remote monitoring strategy was implemented for the sites that could no longer be visited. Methodologies to prevent breaches of confidentiality, protection of data privacy, and to assure blinding were developed. Several years later, during the COVID-19 crisis, we could successfully fall back to the learnings of that period.

The efficacy of a new product assessed in a clinical trial under controlled circumstances and its effectiveness in a real-world setting may be significantly different. Shortcomings in the functioning of the health system and the resulting access restrictions may hamper or completely jeopardize the effect of a theoretically powerful intervention [51]. Therefore, implementation research on the mitigation of this challenge is necessary. This should be considered during the early planning stage and include all the necessary local stakeholders. This was carried out by a carefully planned and conducted Phase IIIb study for fexinidazole in a broader patient population [45].

An additional, specific challenge for HAT is the current change of elimination strategy [52] from the use of mobile teams to passive detection in health centers. It remains to be seen if and how the loss of experience, due to decentralization and declining prevalence, will impact the success of fexinidazole [53].

## 7. Conclusions: Lessons Learnt

The success story of fexinidazole is instructive in several ways [54]. First, it shows that the one molecule that will make it and become a new drug is probably neither the most active of its class nor the most bioavailable or the least toxic (Figure 3). Instead, it will be the molecule that meets all the requirements of the target-product profile in a satisfactory way. Fexinidazole was not an outstanding candidate in any particular aspect (Figure 4), but it effected an acceptable performance in all the relevant tests; thus, setting the bar too high for new drug candidates can be counterproductive [55]. The transition from the preclinical to the clinical development of fexinidazole also underscores the high predictive value of the *T. brucei* mouse models [33,56], demonstrating that a primate model of infection is not absolutely necessary for the development of new drugs against human African trypanosomiasis.

The example of fexinidazole has also shown that efficient drug development in extremely low resource settings is possible. The clinical studies have successfully passed remote regulatory assessment by the US FDA. Clinical trials on neglected diseases limited to remote regions have a different reality, even compared to diseases such as malaria, HIV or tuberculosis, which also occur in low resource countries, but the trials can be carried out in teaching hospitals or excellence centers. It was paramount for the HAT phase II/III trials that all professionals had a fundamental understanding of the importance of the initial and continued investment in staff training, the expenses and efforts to improve and maintain the technology, the logistics, and the need for very detailed and seamless quality control and assurance measures.

Finally, fexinidazole reminds us that drug discovery remains a risky business. As outlined above, the risks can be mitigated by good planning and preparation. Yet despite all precautions, moving to stage II/III clinical development bore a substantial degree of uncertainty. It required champions such as DNDi’s Els Torreele, Nathalie Strub-Wourgaft and Bernard Pecoul, who were willing to take on the risk of re-engaging with a class of molecules that had been abandoned by pharma, to drive the R&D of fexinidazole as the first oral treatment of HAT. The visionary contribution by Marcel Tanner in his dual role as the Director of the Swiss TPH and Chair of the Board of Directors of DNDi cannot be overstated.

## Figures and Tables

**Figure 1 diseases-10-00090-f001:**
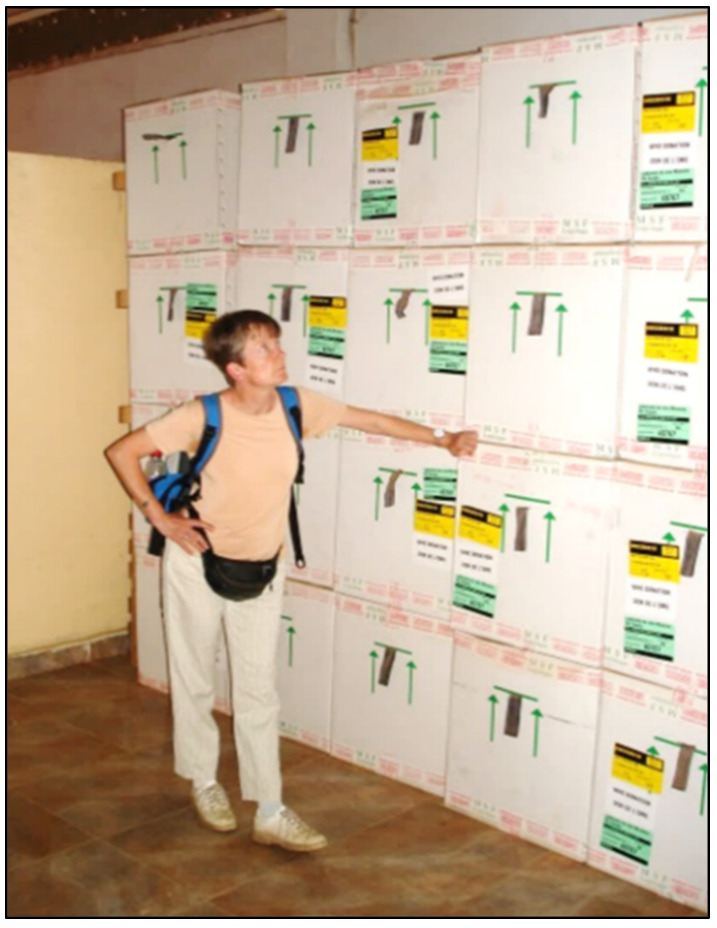
NECT presents a logistic challenge in the remote rural areas where HAT is prevalent. One box on the picture contains the treatment for two patients with eflornithine infusions. Later, the same boxes contained the NECT treatment (nifurtimox tablets and eflornithine infusion bags) for four patients. The picture shows Gabriele Pohlig, a key person for the NECT and DB289 trials, who sadly passed away in June 2022. Photograph by Sonja Bernhard.

**Figure 2 diseases-10-00090-f002:**
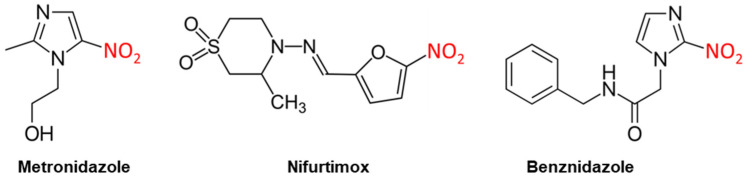
Nitro-drugs are a mainstay of antiprotozoal chemotherapy. The 5-nitroimidazole metronidazole is used for intestinal pathogens and trichomoniasis; the 2-nitroimidazole benznidazole for Chagas disease; the nitrofuran nifurtimox for Chagas and, in combination with eflornithine, for *gambiense* HAT.

**Figure 3 diseases-10-00090-f003:**
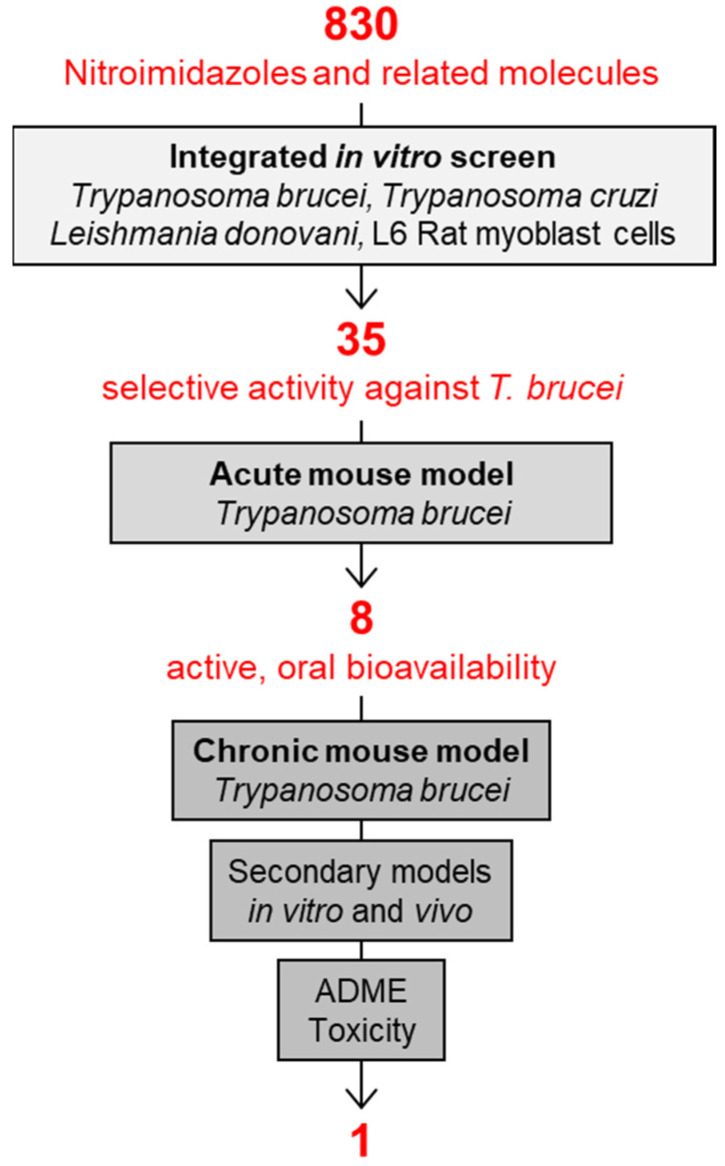
The HAT-pipeline at Swiss TPH established by the Parasite Chemotherapy Unit under the lead of Reto Brun. The one molecule that was singled out as the best drug candidate of all the tested nitro-drugs was fexinidazole.

**Figure 4 diseases-10-00090-f004:**
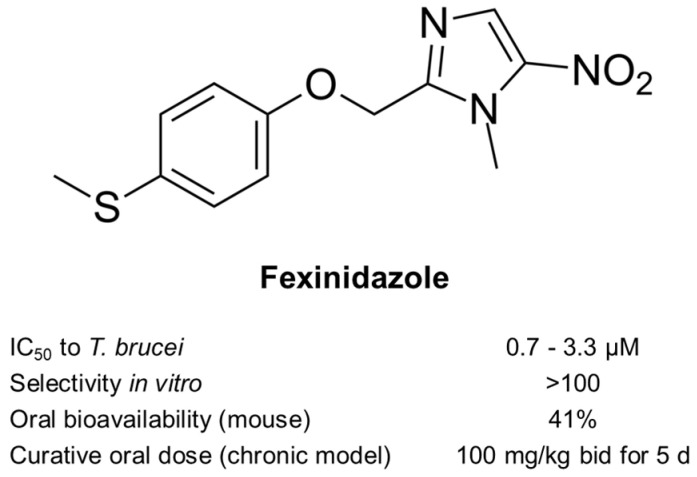
Fexinidazole and its efficacy as assessed in vitro (axenic cultivation of *T. brucei* subspecies bloodstream forms) and in vivo (mouse models of infection). IC_50_, 50% inhibitory concentration; bid, twice daily [31,32].

**Figure 5 diseases-10-00090-f005:**
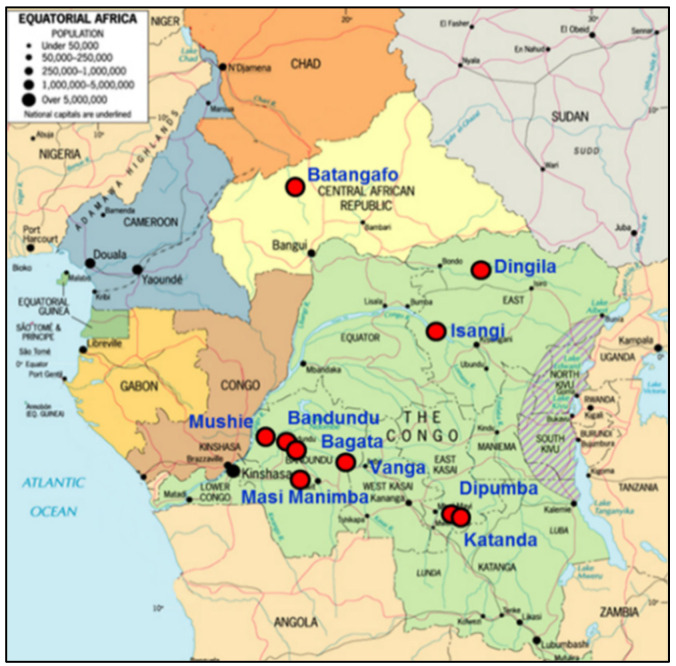
The study sites of the fexinidazole combined phase II/III clinical trial.

**Figure 6 diseases-10-00090-f006:**
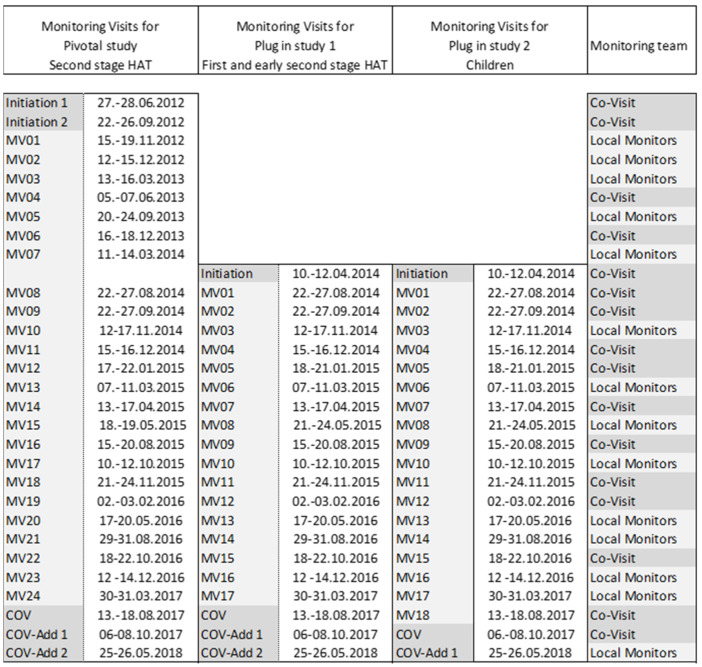
Example of a monitoring visit schedule for one of the high-recruiting study sites (MV, monitoring visit; COV, close out visit; Add, additional visit).

## Data Availability

Not applicable.

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
