# Peer review of "Fexinidazole for Human African Trypanosomiasis, the Fruit of a Successful Public-Private Partnership"

_diseases, 2022, doi:10.3390/diseases10040090_

Round 1

Reviewer 1 Report

Title: Fexinidazole for human African trypanosomiasis, the fruit of a 2 successful public-private partnership.

I recommended that manuscript could be accepted with MINOR MODIFICATIONS. The review is an historical perspective about fexinidazole development; however, if available scientific data could be extended, manuscript will be better.

-          In abstract, include some quantitative results obtained for fexinidazole, in preclinical and clinical studies.

-          In the preclinical studies, a table with in vitro and in vivo results could be easier to lectors according the different tested species of parasites. In both cases compare with reference drug in each study. Same for clinical studies reported in literature.

-          Include more description related with: mechanism of action, toxicity and pharmacokinetic properties.

-          Patents retrieved from this drug could be mentioned, as well as cost of treatment.

Reviewer 2 Report

It is an interesting review on the drug discovery of fexinidazole to treat HAT. The paper is written in a straightforward manner and is easy to read. There are minor points maybe improve its quality. The part of drug history may be concentrated in a table containing each drug and its side effect. Sometimes the authors begin the sentence with numbers like in line 108 they started with 80, it is favorable to write Eighty. The authors didn't mention the effect of fexinidazole on the liver and kidney. 

The review needs a conclusion. Lastly, what is the commercial price of the drug? 
